# Application of PSO-BPNN-PID Controller in Nutrient Solution EC Precise Control System: Applied Research

**DOI:** 10.3390/s22155515

**Published:** 2022-07-24

**Authors:** Yongtao Wang, Jian Liu, Rong Li, Xinyu Suo, Enhui Lu

**Affiliations:** 1State Key Laboratory of Advanced Design and Manufacture for Vehicle Body, Hunan University, Lushan South Road, Yuelu District, Changsha 410082, China; yylirong@hnu.edu.cn (R.L.); suoxinyu@hnu.edu.cn (X.S.); 2Guizhou Institute of Water Resources Science, Guiyang 550002, China; 3School of Mechanical Engineering, Yangzhou University, Yangzhou 225012, China; luenhui@hnu.edu.cn

**Keywords:** PSO-BPNN-PID, nutrient solution EC regulation, wireless sensor network acquisition device, simulation and experiments

## Abstract

In this paper, we present a nutrient solution control system, designing a nutrient solution electrical conductivity (EC) sensing system composed of multiple long-range radio (LoRa) slave nodes, narrow-band Internet of Things (NB-IoT) master nodes, and a host computer, building a nutrient solution EC control model and using the particle swarm optimization (PSO) algorithm to optimize the initial weights of a back-propagation neural network (BPNN). In addition, the optimized best weights are put into the BPNN to adjust the proportional–integral–derivative (PID) control parameters Kp, Ki, and Kd so that the system performance index can be optimized. Under the same initial conditions, we input EC = 2 mS/cm and use the particle swarm optimization BP neural network PID (PSO-BPNN-PID) to control the EC target value of the nutrient solution. The optimized scale factors were Kp = 81, Ki = 0.095, and Kd = 0.044; the steady state time was about 43 s, the overshoot was about 0.14%, and the EC value was stable at 1.9997 mS/cm–2.0027 mS/cm. Compared with the BP neural network PID (BPNN-PID) and the traditional PID control approach, the results show that PSO-BPNN-PID had a faster response speed and higher accuracy. Furthermore, we input 1 mS/cm, 1.5 mS/cm, 2 mS/cm, and 2.5 mS/cm, respectively, and simulated and verified the PSO-BPNN-PID system model. The results showed that the fluctuation range of EC was 0.003 mS/cm~0.119 mS/cm, the steady-state time was 40 s~60 s, and the overshoot was 0.3%~0.14%, which can meet the requirements of the rapid and accurate integration of water and fertilizer in agricultural production.

## 1. Introduction

Precision irrigation and precise fertilization in the agricultural production process can not only improve the yield and quality of agricultural products but can also effectively solve the environmental problems caused by excessive fertilization, which is in line with the concept of ecological green development. Generally, the suitable nutrient solution EC range for crops is usually between 0.8 mS/cm and 2.5 mS/cm, and the EC error range is ±0.8 mS/cm. However, the regulation process of EC in nutrient solutions has the characteristics of nonlinearity, time delays, and time variations. The traditional PID controller, suitable for a simple linear system, will have a large error in the EC regulation of a nutrient solution. Therefore, a large number of studies on EC regulation of nutrient solutions have been carried out in several countries, mainly focusing on combinations of fuzzy control and PID, heuristic algorithms and PID, etc., as follows.

(1)Combination of fuzzy control and PID: Wang Xiaolong designed a fuzzy PID controller to control the precise ratio of water and fertilizer [1]. Zhang Yubin et al. applied fuzzy PID control technology based on the EC value and pH value and developed a precise water and fertilizer irrigation control system [2]. Wang Haihua et al. adopted a PI and fuzzy subsection control strategy to better solve the lag and instability problems of water and fertilizer EC regulation [3]. Li Li et al. analyzed the situation of an actual closed cultivation system. To meet their requirements, the structure of the nutrient solution to control the secondary mixed fertilizer was designed, and a mathematical model of the dynamic process of the nutrient solution was established. At the same time, a PI control algorithm was also designed, and their testing verified that the steady-state time dimension of the system was 100 s, the overshoot was 3%, and the control performance was excellent [4].(2)Combination of heuristic algorithm and PID: the BP neural network has a strong nonlinear mapping ability and self-adaptive ability, and its self-learning ability can be used to output the optimal PID controller parameter combination corresponding to a certain optimal control and thus achieve the desired control effect. However, the initial value of the weight of the traditional BP neural network is randomly selected according to experience. This method leads to slow convergence of the network, which makes it easy to fall into the local optimal solution, and the final result has a large degree of instability [4]. Because the particle swarm algorithm has memory, it can transfer the memory of the best position of the particle in the history of the group to other particles and has the advantages of having fewer parameters to be adjusted and a simple structure, so the particle swarm algorithm was selected to weight the BP neural network. Optimization was performed in the design of PSO-BPNN-PID to improve the control effect. In recent years, scholars have carried out a series of studies in this area. Jiang Liu et al. designed a PID controller based on the BP neural network algorithm and analyzed the vehicle dynamics. The simulation results showed that the dynamic performance of the vehicle was effectively improved under different input conditions [5]. In order to reduce the influence of temperature on the micro-gyroscope, Xia Dunzhu et al. proposed a temperature compensation control method. First, a BP (back-propagation) neural network and polynomial fitting were used to build the temperature model of the micro-gyroscope. Considering the requirements of simplicity and real-time performance, piecewise polynomial fitting was adopted in the temperature compensation system. Then, an integral-separated PID temperature control system was adopted, with a proportional–integral–derivative control algorithm to stabilize the internal temperature of the micro-gyroscope and achieve its optimum performance. The experimental results showed that the combined temperature compensation and control method of the micro-gyroscope could be realized effectively in a prototype of the micro-micro-gyroscope [6]. Alex Alexandridis et al. proposed a new approach to controlling the general properties of nonlinear systems, using an inverse radial basis function neural network model that was able to combine disparate data from a variety of sources. The results revised the ability of the proposed control scheme to process and manipulate a variety of data. Through the data fusion method, it was shown that the method responded in a faster and less oscillatory manner [7]. Jun Wang et al. proposed a closed-loop motion control system based on a BP neural network (BPNN) PID controller, which used a Xilinx field programmable gate array (FPGA) solution. The results showed that the proposed system could realize self-tuning PID control parameters and had the characteristics of reliable performance, high real-time performance, and strong anti-interference ability. Compared with MCU, the convergence speed was far more than three orders of magnitude, proving its superiority [8]. Yuan Jianping [7] used the GA-PSO-BP-PID algorithm to control the greenhouse environment, and the simulation results obtained using MATLAB showed that the stability and robustness of the control system were better than conventional BP-PID [9]. Li Hang et al. used an improved genetic algorithm to optimize the BP neural network to achieve better control over the gas concentration [10]. The abovementioned BP neural network research has achieved good results in the field of environmental control but not in the field of nutrition. Therefore, based on the above research, this paper develops an EC control system for a nutrient solution consisting of a sensing layer, a network layer, and an application layer. The sensing layer consists of multiple LoRa sub-nodes and one LoRa master node, where the master node is connected to the water and fertilizer integration system. By combining LoRa and NB-IoT to form a wireless sensor network, comprehensive sensing, reliable transmission, and the intelligent application of water and fertilizer control information are realized. The nutrient solution EC regulation model of the water–fertilizer integration system is further constructed, and the PSO-BPNN-PID controller is designed by combining PSO optimization with the BP-PID coupling control method, and the initial weights of BPNN are continuously optimized by the PSO algorithm to achieve the optimal weights, and the optimal weights are input into BP neural network to automatically adjust the PID control parameters Kp, Ki, and Kd to find the optimal control parameters. After a MATLAB simulation with a good control effect, further, through the self-organized network communication performance test and nutrient solution EC control system test, it is proved that the nutrient solution EC control system has excellent performance and can meet the needs of actual production.

## 2. Working Principle of the Integrated Water and Fertilizer Device

For nutrient solution regulation, researchers mostly adopt water and fertilizer integration methods. Based on the analysis of nutrient solution regulation devices in China and abroad, in this study, we have designed a multi-channel water and fertilizer integration device, as shown in Figure 1. The device is mainly composed of seven parts: ① an industrial aluminum profile frame, ② the main network system, ③ the fertilizer suction system, ④ the mixed fertilizer system, ⑤ the detection system, ⑥ the control system, and ⑦ the fertilization system. This system realizes the automatic and intelligent adjustment and control of irrigation and fertilization according to input conditions or sensor data, such as soil moisture, evaporation, rainfall, and light intensity. Automatic timing and quantitative irrigation and fertilization of crops in irrigated areas.

### 2.1. Working Principle

The preparation process for the nutrient solution is as follows: (1) The water source 19 is continuously injected into the buffer mixing tank 2 at the flow rate *Qw* through the pressurized pipeline, whereas the fertilizer pump 1 extracts the nutrient solution from the bottom of the buffer mixing tank 2, and a part of the nutrient solution passes through the connection. After the pipeline passes through the Venturi jet 3, *Q_F_* is injected into the buffer mixing tank 2 again. Another part of the nutrient solution *Qout* enters the irrigation area through the pipe network for fertilization. (2) Under the action of the venturi effect, the fertilizer mother liquor *Q_NS_* is continuously sucked into the Venturi jet. After passing through the Venturi jet 3, the *Q_F_* is finally injected into the buffer mixing tank 2. The fertilizer mother liquor *Q_NS_* and water *Qw* are mixed for the first time in the Venturi jet 3 and are mixed for the second time in the buffer mixing tank 2 to form a two-time mixing mode. The uniformity and preparation efficiency of the fertilizer mother liquor, *Q_NS_,* and water, *Qw*, are greatly improved. The working principle of the integrated water and fertilizer system is shown in Figure 1 [11,12].

### 2.2. Controller Selection

For the human–machine interface of the controller, we adopted the Kunlun Tong-state TCP1031NI. The core controller is an intelligent IoT touch screen with Cortex-A7 multi-core CPU, main frequency 1GHz, equipped with a 10.1-inch TFT LCD screen, pre-installed McgsPro configuration software (running version), Serial interface: RS232/RS485, Ethernet port: 1 × 10/100 M adaptive. The fertilizer spreader controller mainly includes the main interface, manual operation mode, automatic operation mode, etc.

## 3. Design of EC Regulation System of Nutrient Solution

The system can be divided into three parts, consisting of the sensing, network, and application layers. The network structure of the EC regulation system of nutrient solution is shown in Figure 2.

### 3.1. Sensing Layer

The sensing layer consists of multiple LoRa slave nodes and one NB-IoT master node. Each slave node has the ability to collect 4–20 mA, 0–5 V, and 485 standard signals of pressure and water level transmitters; to collect flow data from flow meters and pulse meters for cumulative flow calculation; to collect on-site signals such as ambient temperature and humidity and transmit them through LoRa to the master node.

Each slave node is powered by a 12 V high-performance lithium battery and is powered by a standard power adapter. It has a built-in 5 V voltage regulator chip and a 3.3 V voltage conversion circuit, which can collect the output signals of various instruments and transmitters and uses LoRa for long distances. It transmits data, controls the operation of a 12 V solenoid valve (switch value), a DC3.6V motor (analog value), collects motor operation status, and transmits data remotely through an NB-IoT signal, which is suitable for monitoring sites without power supply conditions and in harsh environments. Since the node needs to work for a long time in the field, a photovoltaic solar charging device was designed, which has the functions of undervoltage protection, overdischarge protection, overload protection, floating charging settings, overload short circuit protection, overvoltage protection, direct charging voltage, and other functions. Figure 3 shows the structure of the slave master node.

The master node is mainly based on the STM32 controller and is connected to the network through modbus and UDP (User Data Protocol) communication to ensure the reliability of the data and instructions. Through the multi-node coverage and high timeliness of the system, the centralized analysis of data in the irrigation area, as well as remote control and scheduling in the gate cluster, are realized, and it is suitable for the precise control of water delivery and distribution in small and medium-sized channels for farmland irrigation. At the same time, the master node is connected to the water and fertilizer integration system.

#### 3.1.1. Design and Implementation of LoRa Wireless Communication

For LoRa, we adopted the E22-400T22S wireless serial port module and multiple LoRa slave nodes in a wireless network with a single NB-IoT master node controller LoRa. The LoRa spread spectrum wireless serial communication module, with an operating frequency of 433 MHz, and a power of 1000 mW, was used here. This module is an industrial-grade wireless serial port module with high stability. We adopted Semtech’s latest generation of LoRa spread spectrum design and development, and LoRa spread spectrum modulation. The measured transmission distance can reach up to 8000 m. The module has more transmission methods and supports data broadcasting, data monitoring, fixed-point transmission, master-slave mode, automatic relay, fixed-point wake-up, automatic response, and other transmission methods. More comprehensive functions include support for ultra-low power consumption, wireless configuration, group package configurability, a package separator, output address, and other functions. Due to these diversified functions and high stability, the system can be widely used in various environments to easily realize low-frequency wireless data transmission.

#### 3.1.2. NB-IoT Master Node Communication with NB-IoT Child Nodes

(1)LoRa transmission configuration

LoRa is divided into a master and slave structure. The master can send data to the specified slave, and all of the slaves can receive the data sent by the master, as shown in Table 1. At this time, the steps undertaken by the module to set the master–slave mode are as follows: (1) MD0 = 0, MD1 = 0; MD0 = 0, MD1 = 1. (2) It is necessary to configure the 0x07 special function register to 0x0004, master-slave mode. (3) The sending module and the address of the receiving module can be set differently. (4) The channels of the sending module and the receiving module are set to be the same. Any module sends data, and the specified module can receive it. When multiple receiving modules have the same address channel, they can all receive data. The command to set the module fixed-point transmission is as follows. (5) MD0 = 0, MD1 = 0; MD0 = 0, MD1 = 1. (6) The special function register 0x07 needs to be configured to 0x0002, fixed-point transmission mode. (7) The address of the sending module and the receiving module can be set to be different. (8) The channels of the sending module and the receiving module can be set to be different.

(2)LoRa communication protocol

The child node (front-end data acquisition controller) communicates with the master node LoRa, the master node (channel 18, group number address {0xFF 0xFF}), and the child node (channel 18, group number address {0xXX 0xXX}) in the same channel. The child node sends frame data in the following format: 0xFF 0xFF 0x41 0x42 0x31 0x31 0x31 0x30 0xE8 0x0b 0x0a 0x00 0x. Table 1 provides information on master node–child node protocol nesting.

The master node obtains data by sending a request to the slave node. When the master node sends a request to the slave node, it needs to add the wake-up code FF FF FF FF FF FF FF FF FF FF (10 0xFF), and the wake-up code is sent first, with an interval of 0.2~0.5. After this, the request frame is sent again. The protocol frame format can be described as follows.

The frame is the basic unit for transmitting the information. Each frame is composed of seven fields, including the frame starter, address field, control field, data length field, data field, frame information check field, and frame terminator. The format is shown in Table 2.

The frame start character (STA) identifies the beginning of a frame of information, 1 byte, and its value is fixed at E8H = 11101000B. The address field (AD) is the address that identifies the current receiving (sending) device, two bytes, and 0xFFFF is the broadcast address, which is the address of the LoRa transmission channel. (C) is the control field control code indicating the required operation, two bytes, and the format is shown in Table 3.

D7 indicates the transmission direction. D7 = 0 indicates the command frame sent by the master node (repeater). D7 = 1 indicates the response frame sent by the front-end collector. D6 represents the abnormal flag. D6 = 0 indicates the correct response. D6 = 1 indicates the response to the abnormal information.

(3)System software design

① Keil program compilation tool

The μVision Keil 5 system provides a complete development solution, including a powerful emulation debugger that includes a macro assembly, a C compiler, library management, and a linker. They are all put together in an integrated development environment (MDK5). It can perform online debugging, as well as low-level assembly language and advanced computer language C language programming, so this development was completed using the Keil integrated development environment.

② System task design

The system was divided into one main task and six sub-tasks. The main task mainly consists of communication with the upper computer, and through continuous verification, the system receives the instructions of the upper computer and completes the relevant tasks. The subtasks mainly consist of the collection and storage of data on the nutrient solution EC, crop EC limit settings, EC data transmission, switch fertilization valve, fertilization decision, watchdog timer, etc. The overall function of the lower computer is completed through the rotation between tasks. The system task diagram is shown in Figure 4.

### 3.2. Network Layer

The background obtains data by sending a request to the sensing layer (master node). (1) The master node subscribes to the “Topic” background device topic (master node 1 subscribes to the background device topic: /D001), and the master node 1 publishes the “Publish” device topic (master node 1 publishes the device topic:/publish/topic/D001). (2) The background subscribes to the topic of the Topic master node 1 device (master node 1 publishes device topic: /publish/topic/D001), and the background publishes the Publish background device topic (backend publishes device topic: /D001).

### 3.3. Application Layer

The application layer data monitoring and the interface display and supports data communication with the multi-center. For the application layer database, we adopted MySQL, which has different system functions for different roles and adopted the E-R description structure model to design and describe E-R, such as user information, role, menu-role, information collection parameters, and equipment control, among which E-R information such as user, role, and menu is used. The corresponding part of the relationship model shown in the Figure is as follows: (1) User (ID, name, password, role ID, whether to enable, creation time, last login time, extended field 1, extended field 2, extended field 3). (2) Menu bar (menu ID, menu name, menu path, parameter setting, whether to enable, creator ID, creation time, last modification time, extended field 1, extended field 2, extended field 3). (3) Role (role ID, role name, whether to enable, valid time, expiration time, remarks, creator ID, creation time, last modification time, last modification code, extended field 1, extended field 2, extended field 3). (4) Menu-role relationship table (relationship ID, role ID, menu ID, valid time, expiration time, remarks, creator ID, creation time, last modification time, extended field 1, extended field 2, extended field 3). (5) Collection information parameters (collection information ID, device ID, sampling time, sensor number, parameter value). (6) Control information (control information ID, device ID, sensor number, control parameters, sending time). (7) Device-to-belonging irrigation area relationship table (device ID, Irrigation District ID).

## 4. Nutrient Solution System Control Model

Since the Venturi jet mixes the fertilizer while absorbing fertilizer, it can be considered that the virtual mixing volume (VF) is added into the buffer mixing barrel 2. The mixing process is a combination of piston flow (plug flow), and ideal stirred mixing, and the whole system is a second-order time-delay system. *Q**w* = 1 L/s, *Q**_F_* = 0.05 L/s, *Q**_NS_* = 0.01 L/s. The volume of the fertilizer mixing drum is 10 L (the effective working volume is 9 L), and the transfer function of the system is expressed as in Equation (1), where *V**_T_* = 5 L and *V**_F_* = 4 L.
(1)$$\begin{array}{l} \frac{E_{C}(S)}{Q_{NS}(S)}=Gp(S)e^{-\tau s}=\frac{K_{2}}{\left(T_{F}S+1)(TpS+1\right)}e^{-\tau ’s}=\frac{0.95}{320S^{2}+84S+1}e^{5s}\\ K_{2}=\frac{1}{\left(Q_{F}+Q_{W}\right)}=0.95\\ T_{F}=\frac{V_{F}}{Q_{F}}=80\\ Tp=\gamma_{1}\cdot Tr=\frac{4}{5}\cdot \frac{V_{T}}{Q_{W}}=0.8\cdot 5=4\\ \tau =(1-\gamma_{1})\cdot Tr=1 \end{array}$$
where *E**_C_*(*S*) is the output value of the desired nutrient solution conductivity (mS/cm); *Q**_NS_*(*S*) is the output value of the fertilizer mother liquor flow (L/s); *K*_2_ is the second-order system gain (L/s) resulting from the addition of the venturi jet; *T_F_* is the time constant of the nutrient solution preparation process in the Venturi jet (s); *V**_F_* is the increased premixing volume (L) resulting from the addition of the Venturi jet; *γ*_1_ is the mixing coefficient (in this paper *γ*_1_ = 0.8, whereas *γ*_1_ = 0 for the advection mode and *γ*_1_ = 1 for the ideal stirring mode); *T**r* is the delay time of the system (s); *V**_T_* is the effective mixing volume of the mixing barrel (L); *Q**_F_* is the liquid flow into the Venturi jet (L/s); *Q**_W_* is the water flow into the mixing barrel (L/s); *τ* is the lag of the fertilizer mother liquor mixing in the buffer mixing barrel and measurement lag time, including the flow time of the liquid in the pipe and the mixing time (s); *τ’* is the new lag time (s); and *Q**_NS_* ≤ *Q**_F_* < *Q**_w_*, *Q**_NS_* is the flow of the fertilizer mother liquor injected into the fertilizer barrel (L/s) [13,14,15].

## 5. PSO-BPNN-PID Control Model

### 5.1. PSO-BPNN-PID Structure

The PSO-BPNN-PID controller consists of two parts, namely, the PSO-BPNN and the conventional PID controller. According to the error e between the input r and the output y and the three parameters Kp, Ki, and Kd, the conventional PID controller obtains the controller output u through the control algorithm and then obtains the system output Y_out_ from the transfer function of the control object, thereby realizing PID. The controller directly controls the closed-loop control of the controlled object. As long as the system is running, the BP neural network can adjust and optimize the weighting coefficient through its own self-tuning ability and automatically adjusts the PID control parameters to obtain the optimal control parameters and cause the system performance indicators to reach the ideal state [16,17]. The structure of the PSO-BP-PID controller is shown in Figure 5. The nutrient solution EC-controlled BPNN structure is shown in Figure 6.

### 5.2. PSO-BPNN-PID Algorithm

The initial weights of BPNN are continuously optimized by means of the PSO algorithm to obtain the best weights, and the best weights are input into the BP neural network for self-learning adjustment [18].

**Step 1:** Established BP neural network.

(1) Determine the BP neural network structure and select the learning rate *xite* and inertia factor *alfa*. The network adopts a three-layer network architecture with one input layer, one hidden layer, and one output layer. The number of nodes in the input layer of the BP neural network is determined by trial and error. The input vector *x* = [*x*_1_, *x*_2_, *x*_3_, *x*_4_], where *x*_1_ = *e*_k_ − *e*_1_; *x*_2_ = *e*_k_; *x*_3_ = *e**_k_*_−2_ · *e*_1_ + *e*_2_; *x*_4_ = 1. The number of hidden layer nodes is h = 5, and the output layer O = 3, corresponding to the outputs kp, ki, and kd. Therefore, the neural network structure constructed in this paper was 4-5-3 [19,20].

Since the activation function sigmoid of the hidden layer neuron maps the output to [0,1], but the three parameters of the PID controller actually need to be adjusted in a larger range, the array M = [m_1_, m_2_, m_3_] is set as the output gain of the network. The experiments proved that the performance indexes of the system differed when M was taken at different values [21,22]. The experimental results are shown in the table below, and considering each performance index of the system comprehensively, the output gain M = [87, 0.1, 0.96] [23,24].

(2) Initialize the matrix of the input and output weights of the BP neural network. Since the initial weight of the neural network plays a great role in the control effect, the initial value of the weight of each algorithm is different. Among these, the initial weight of the BP-PID controller is a random number in the interval [−1,1]. The initial weight wij of the PSO-BP-PID controller is determined by trial and error as 4 × 5, and wki is 3 × 5 [25,26]. The initial weight of the controller is shown in Table 4.

**Step 2:** PSO optimization.

(3) Initialize *N* subgroups. Initialize the particle swarm by specifying that the population size *N* of PSO is 20, the maximum number of iterations *M*^max^ is 50, the inertia coefficient *w* uses the linear weight decay method (2), the particle dimension *D* is 3, the learning factor c_1_ = c_2_ = 0.5, the particle position *x* takes on a range of [−5,5], and the particle velocity *v* takes on a range of [−5,5] [27].
(2)$$w=w^{\max}-(t-1)\cdot \frac{\left(w^{\max}-w^{\min}\right)}{\left(M^{\max}-1\right)}$$

In Equation (2), *t* is the number of iterations, *w*^max^ is 0.9, and *w*^min^ is 0.4.

(4) Construct the fitness function. The square sum of error (*tol_fitness*) is chosen as the fitness function. According to the design flow, the system transfer function for the test is selected as shown in Equation (3).
(3)$$tol\_fitness=tol\_fitness+abs(error{\left(k\right)}^{2})$$
where error(*k*) is the systematic error.

(5) Suppose that there are *N* particles in *D*-dimensional space, $x_{i}=(x_{i1},x_{i2},\cdots x_{id})$ represents the speed of particle i, $pbest_{id}=(p_{i1},p_{i1}\cdots ,p_{id})$ represents the best placement experienced by the individual particle i. Then, $pbest_{d}=(g_{1},g_{2}\cdots ,g_{d})$
represents the best position experienced by the population. In every iteration, the d dimensionality velocity of particle i is updated according to Formula (4):(4)$$v_{id}^{k+1}=\omega v_{id}^{k}+c_{1}\gamma_{1}(pbest_{id}-x_{id}^{k})+c_{2}\gamma_{2}(gbest_{d}-x_{id}^{k})$$

The d dimensionality position of the particle i is updated according to Formula (5):(5)$$x_{id}^{k+1}=x_{id}^{k}+v_{id}^{k+1}$$
where $V_{id}^{k}$ represents the d dimensionality component of the flight velocity vector of particle i in the k iteration, and $x_{id}^{k}$ represents the d dimensionality component of the position vector of particle i in the k iteration. $c_{1}$ and $c_{2}$ indicate that the learning factor is a non-negative constant, with a value range of [1.2, 2]. $c_{1}$ is used to adjust the step size of particles flying to the optimal position and $c_{2}$ is used to adjust the step size of particles flying to the optimal position of the whole group. $\gamma_{1}$ and $\gamma_{2}$ represent random numbers in the interval of [0,1], and ω is the inertial weight, which describes the influence of particle inertia on velocity; its value can be used to adjust the global and local optimization ability of particle swarm optimization. $-v^{\max}\leq v_{id}^{k+!}\leq v^{\max}$, $v^{\max}$ is a predetermined normal number limiting the speed range. According to the specific problem, the iteration termination condition is generally selected as the maximum iteration number, or the optimal position searched so far by the particle swarm, and it meets the predetermined minimum adaptive threshold [28].

(6) The particle fitness value is calculated by means of Formula (3), and a judgement is made as to whether the termination condition is satisfied (the maximum number of iterations is reached). If so, the parameter optimization ends, and the algorithm exits; otherwise, it jumps to (5).

(7) Compare the historical optimal position of the population with the current optimal particle position. If the current search individual is better, update and replace the historical optimal position of the population, update the historical optimal particle position *x* and velocity *v*, and use Formula (6) to obtain the initial values of the input and output weights after PSO optimization, which are expressed as *wi^init^* and *wo^init^*.
(6)$$\begin{array}{l} wi^{init}(t,:)=x\;(1,\;(t-1)\cdot m:t\cdot m)\\ wo^{init}(r,:)=x\;(1,\;(m\cdot q+1)+(r-1)\cdot q):\;((m\cdot q+1)+r\cdot q-1)) \end{array}$$

**Step 3:** BP neural network training.

(8) The input of the neural network is obtained by sampling. For the activation function from the input layer to the hidden layer, we adopt a sigmoid function, and for the activation function from the hidden layer to the output layer, we adopt a non-negative sigmoid function; see Formula (9) for the calculation process. Via forward propagation, one can calculate the error [29].

The input of the *j*-th neuron in the input layer is *x*_j_ and the output is(7)$$O_{j}^{\left(1\right)}=xj,\;j=1,2,3,4$$

The input of the *i*-th neuron in the hidden layer is
(8)$$net_{i}^{\left(2\right)}(k)=\sum_{j=1}^{4}w_{ij}^{\left(2\right)}O_{j}^{\left(1\right)},i=1,2,3,4$$
where $w_{ij}^{\left(2\right)}$ is the connection weight between the *i*-th neuron in the hidden layer and the *j*-th neuron in the input layer.

The output of the *j*-th neuron in the hidden layer is
(9)$$O_{i}^{\left(2\right)}=f(net_{i}^{\left(2\right)}(k))$$
where f (⋅) is the activation function; here, the sigmoid activation function is used:(10)$$f(x)=\frac{e^{x}-e^{-x}}{e^{x}+e^{-x}}$$

The input of the *l*-th neuron in the output layer is
(11)$$net_{l}^{\left(3\right)}(k)=\sum_{j=1}^{5}w_{lj}^{\left(3\right)}O_{j}^{\left(2\right)},i=1,2,3$$
where $w_{li}^{\left(3\right)}$ is the connection weight between the *l*-th neuron in the output layer and the *i*-th neuron in the hidden layer.

The output of the *l*-th neuron in the output layer is
(12)$$O_{l}^{\left(3\right)}=g(net_{l}^{\left(3\right)}(k))$$
where $O_{1}^{\left(3\right)}=K_{p}$, $O_{2}^{\left(3\right)}=K_{i}$, $O_{3}^{\left(3\right)}=K_{d}$. Because *K**_p_*, *K**_i_*, and *K**_d_* are always positive, a non-negative sigmoid activation function is used:(13)$$g(x)=\frac{e^{x}}{e^{x}+e^{-x}}$$

(9) Error back-propagation starts from the output layer to calculate the output error of each layer of neurons layer by layer and then updates the weights according to the error gradient descent method. If the conditions are met, the learning process ends, and *K**_p_*, *K**_i_*, and *K**_d_* are obtained. Otherwise, the system returns to Step 7.
(14)$$E(k)=\frac{1}{2}{\left(r(k)-y(k)\right)}^{2}$$

**Step 4:** PID control.

The outputs *u*(*k*) and *y**_out_*(*k*) are calculated using Formulas (15) and (16). Figure 4 presents a flowchart of the PSO-BPNN-PID algorithm [30,31]. Figure 7 presents a flow chart of the PSO-BPNN-PID algorithm.
(15)$$\begin{array}{c} u(k)=kp\cdot x1+ki\cdot x2+kd\cdot x3\\ yout(k)=-d2\cdot y1-d3\cdot y2+n2\cdot u(k) \end{array}$$

## 6. Simulation and Experimentation

MATLAB simulation analysis was carried out using BPNN-PID, PSO-BPNN-PID, and conventional PID. Setting *r*(*k*) = 2, the controller outputs the initial value *u*(*k*) = 0, and the transfer function is as in Formula (1), taking a sampling time *t*_s_ = 0.01 s, learning rate *xite* = 0.5, inertia factor *alfa* = 0.05 for the simulation experiment. Based on the comparison of different control results shown in Figure 8 and the control error curve of the PSO-BP-PID controller shown in Figure 9, it can be seen that the steady-state time of BPNN-PID control was about 50 s, the overshoot was about 0.15%, and the system’s EC value was stable at 1.9967 mS /cm~2.0030 mS/cm. For PSO-BPNN-PID, the control steady state time was about 43 s, the overshoot was about 0.14%, and the system’s EC value was stable at 1.9997 mS/cm–2.0027 mS/cm. For the traditional PID, the steady-state time was about 130 s, the overshoot was about 21.8%, the system’s EC value was stable at 1.790 mS/cm~2.436 mS/cm, and the oscillation was large. The comparative simulation results showed that the PSO-BPNN-PID method presented in this paper reduced the steady-state time by 16.2% and 67% and reduced the overshoot by 99.3% and 7.1% compared with BPNN-PID and the traditional PID. It demonstrated a faster response speed and higher accuracy. Although the amplitude deviations of the fluctuation of the BP-PID algorithm and the PSO-BP-PID algorithm were not greatly different, it was basically in a coincident state, but in the initial stage of the control process, after the particles were optimized and after the system error function was obtained, the curve was obvious. The convergence speed was accelerated so that the time required for the system to reach stability was significantly reduced.

### 6.1. Analysis of Results

Figure 10 displays the PSO-BPNN-PID fitness curve, and Figure 11 shows the Kp, Ki, and Kd values optimized by means of PSO. It can be seen from Figure 10 that when the number of iterations is about three, the system running time is about 43 s; the fitness value is about 3.9817; the error is less than 10^−3^; the control parameters Kp, Ki, and Kd and the performance indicators of the controller gradually tend to be stable; and the optimal control parameters of the controller are quickly found through the powerful local search ability of the particle swarm algorithm. In the optimization process of particle swarm optimization, the performance index tol_fitness is continuously reduced to reach a stable value, and the control parameters of the controller are constantly approaching an optimal solution. For the unstable nutrient solution EC control system, the control parameters are iteratively optimized by the neural network of particle swarm optimization, and the parameters Kp, Ki, and Kd can be reasonably configured, and thus effective control is achieved. The conventional BP-PID control algorithm is used to control the EC value of the nutrient solution. Since the self-learning of BPNN depends on the selection of the initial weight, the control effect is not satisfactory. The PSO-BP-PID control algorithm uses PSO to control the BP neural network. The thresholds and weights are optimized, each individual calculates the individual fitness value *tol_fitness* through the fitness function and a series of update operations, and the BPNN algorithm continuously updates the weights of the network according to the rate of these optimal individuals. The controller outputs the optimal control parameters. It can be seen from Figure 11 that Kp = 81, Ki = 0.095, and Kd = 0.044 after POS optimization.

With further input EC values of 1 mS/cm, 1.5 mS/cm, 2 mS/cm, and 2.5 mS/cm, respectively, the simulation model was verified. It can be seen from Table 2 that the range of fluctuation of EC values was 0.003 mS/cm~0.119 mS/cm, the steady-state time was 40 s~60 s, and the overshoot was 0.3%~0.14%, which can meet the needs of agricultural production by quickly and accurately realizing the optimal combination of water and fertilizer. The requirements for integration, the control parameters, and the performance indicators of different EC values are shown in Table 5.

### 6.2. System Testing

#### 6.2.1. Wireless Sensor Network Acquisition System Test

After actual verification, the system showed good communication effects, accurate temperature and humidity data, and the ability to complete the setting of irrigation parameters and control the actions of the related executive elements easily and quickly. Furthermore, the switching function of manual/automatic mode enables managers to participate in the control system, complementing the man–machine interface and giving full play to the intelligent irrigation function. The system meets the requirements of actual production and has good application and promotion value.

In this study, the startup time, data transmission time, system packet loss rate, and other performance measures of the water conservancy and water-saving irrigation decision-making system were tested separately, and the tests were performed at short distances (within 1 and 10 m) and long distances (1000–1500 m).

The test results are shown in Table 6 and Table 7. In Table 6, the results of the short-distance test are presented. The test data show that the reporting time was set to be uploaded every 1 s, one data report consisted of 13 instances of data, and the number of tests was 10. The system packet loss rate test results, data delay results, and the distance between the terminal node and the gateway during the test are shown in the table. Due to limited space, in this paper, we chose the longest test distance of 1200 m. For each group, we conducted 10 tests. One thousand packets were sent and received continuously.

It can be seen from the test results that the start-up time of the water-saving irrigation decision-making system was relatively fast, around 3.7 s, and the delay in issuing commands was relatively short, around 1.1235 s. The system was stable and could maintain good operating performance at a long distance, and could be effectively used in the EC regulation of nutrient solutions.

#### 6.2.2. Nutrient Solution Regulation Test

A nutrient solution regulation test platform was built to carry out fertilizer absorption and EC regulation test experiments, as shown in Figure 12. Through the positioning of the rotameter, the fertilizer absorption situation of each channel could be visually checked. At the beginning of the test, the first step was to open check valve 11, manual switch valve 14, and irrigation valve 18. The second step was to turn on the main power supply and main water pump 21 and adjust pressure-reducing valve 22 to ensure that the main pressure was maintained at 0.25 MPa. The next step was to turn on fertilizer suction pump 9 and observe the operation of the system. Step 4: After the system was run normally and stably, we directly read the actual amount of fertilizer absorbed by each channel through the rotameter and repeated the test five times to measure the average amount of fertilizer absorbed [30,31,32].

### 6.3. Test Results and Analysis

Table 8 presents a comparison of the four-channel average fertilizer absorption test data and EC regulation data measured during the test. Under the action of the fertilizer pump, the absorption and pressure output of four different types of single-element fertilizer mother liquor were realized, and the average fertilizer absorption amount was 693.25 L/h, which indicated that the expected test effect was achieved. Comparing and analyzing the testing and simulation data in Table 8, it can be seen that the amount of fertilizer absorbed by two fertilizer absorption channels was slightly lower than that of the other fertilizer absorption channels, but the average relative error was only 4%. The amount of fertilizer absorbed by each channel was relatively balanced, and for the overall fertilizer absorption, the effect was better. The control accuracy of the EC value was +0.07, which meets the needs of actual production.

## 7. Conclusions

(1)In this study, we built a precise EC control system platform for nutrient solutions and developed an EC information perception system composed of multiple LoRa slave nodes, NB-IoT master nodes, and a host computer. The system startup time was about 3.56~3.87 s; the command time was approximately 1.1235 s. The command time was thus in the range of 1.122~1.124 s, and the data delay was about 1.50~11.4 s. The system startup time, command issuing time, data reporting time, and data receiving time intervals were all relatively fast, the delay in issuing commands was small, and the data could be reported according to the time required by the user. The system can maintain good running performance.(2)Based on the PSO optimization method, a BP-PID nutrient solution EC control model with a structure of 4-5-3 was constructed. Under the same initial conditions, when the input EC = 2 mS/cm, the PSO-BPNN-PID method was used to control the EC value of the nutrient solution. The optimized PID scaling factors were Kp = 81, Ki = 0.095, and Kd = 0.044; the control steady state time was about 43 s; the overshoot was about 0.14%; and the system’s EC value was stable at 1.9997 mS/cm–2.0027 mS/cm. Compared with BPNN-PID and the traditional PID, we achieved steady-state time savings of 16.2% and 67%; the overshoot was reduced by 99.3% and 7.1%; the system control performance was excellent.(3)When inputting values of 1 mS/cm, 1.5 mS/cm, 2 mS/cm, and 2.5 mS/cm, we used a Smith PSO-BPNN-PID system to verify the system simulation model, respectively. The results showed that the fluctuation range of EC was 0.003 mS/cm cm~0.119 mS/cm, the steady-state time was 40 s~60 s, and the overshoot was 0.3%~0.14%. The average amount of fertilizer absorbed by the four channels measured in the test was 693.25 L/h, and the control accuracy of the EC value was +0.07, which meets the needs of actual production.

## Figures and Tables

**Figure 1 sensors-22-05515-f001:**
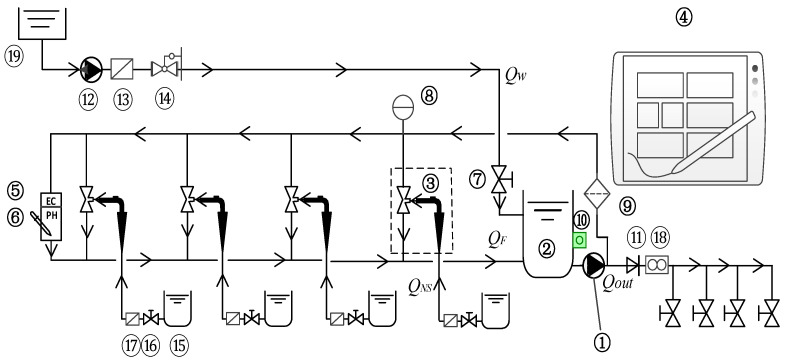
The working principle of the integrated water and fertilizer system. 1. Fertilizer pump; 2. Buffer mixing tank; 3. Venturi jet; 4. Control box (including controller, operation screen and data acquisition module, etc.) system; 5. EC sensor; 6. PH sensor; 7. Float valve; 8. Pressure gauge; 9. Upper branch pipe filter; 10. Pressure-sensitive switch; 11. Check valve; 12. Water source main pipeline water pump (optional); 13. Main pipe filter; 14. Pressure reducing valve; 15. Fertilizer storage tank; 16. Fertilizer injection valve; 17. Butterfly filter; 18. Flow meter; 19. Water source.

**Figure 2 sensors-22-05515-f002:**
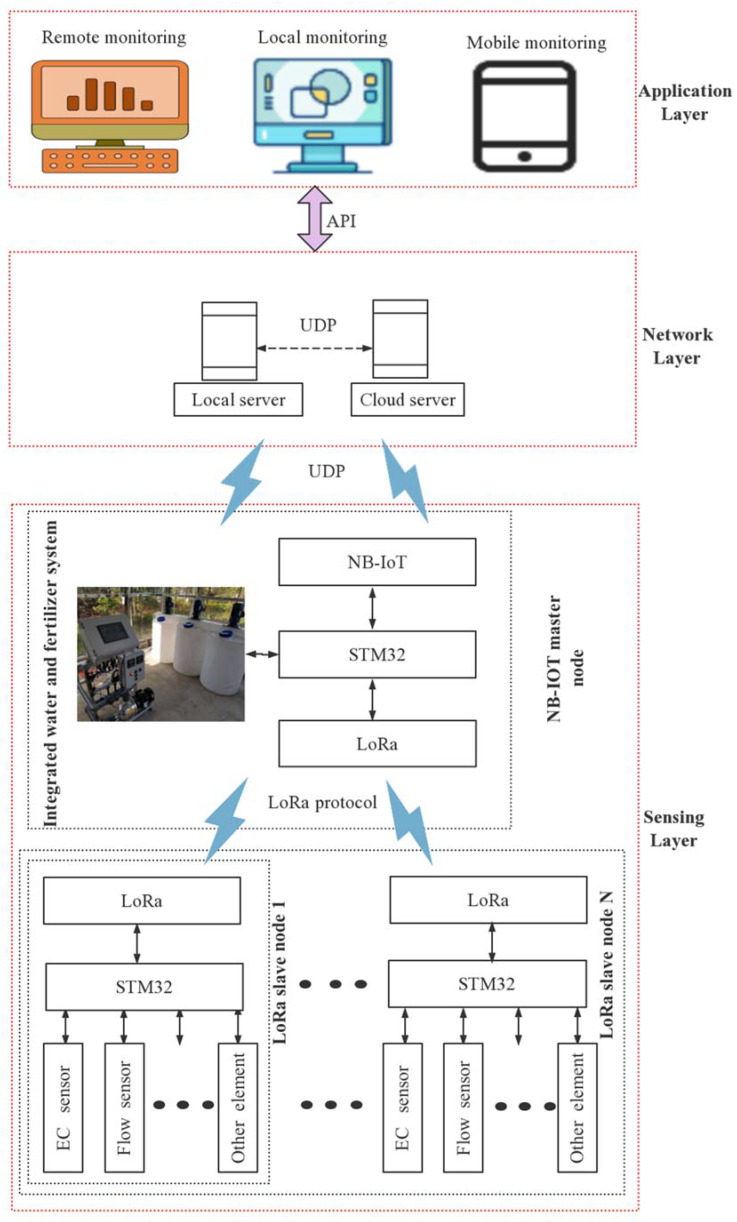
Network structure of EC regulation system of nutrient solution.

**Figure 3 sensors-22-05515-f003:**
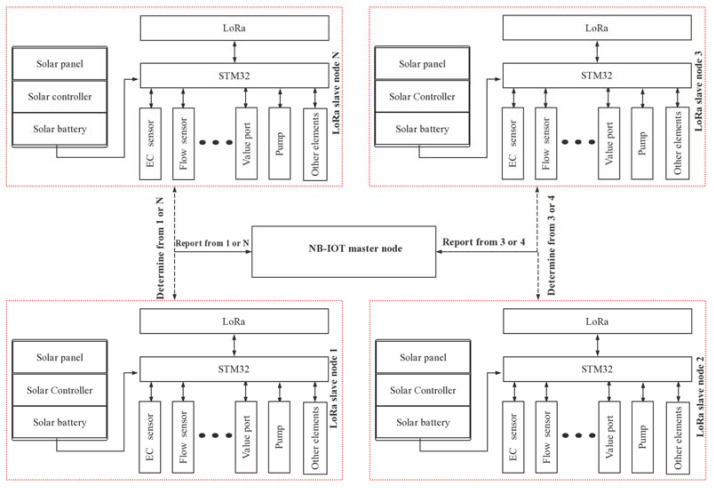
Multi-node LoRa communication structure diagram.

**Figure 4 sensors-22-05515-f004:**
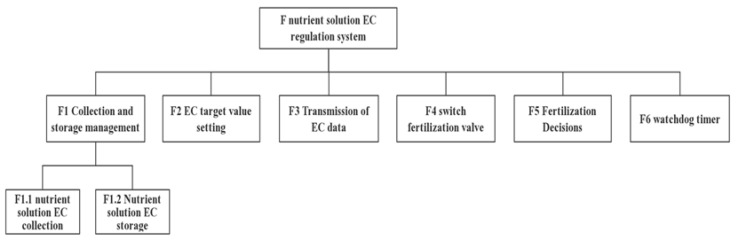
System task diagram.

**Figure 5 sensors-22-05515-f005:**
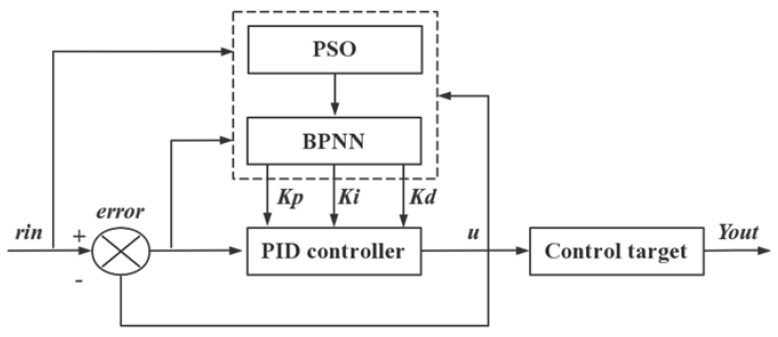
PSO-BP-PID controller structure.

**Figure 6 sensors-22-05515-f006:**
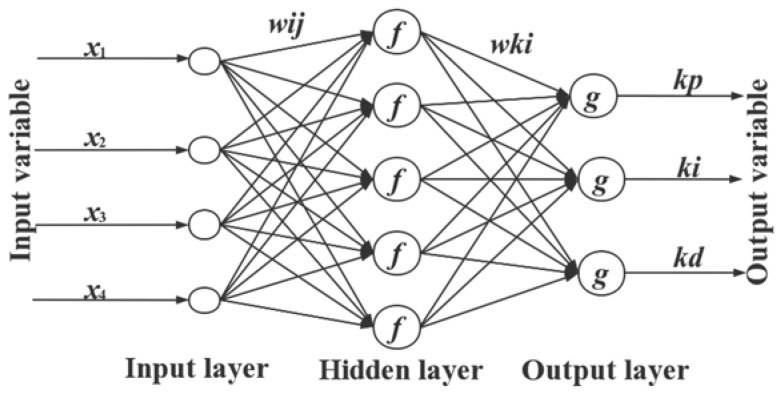
Nutrient EC control BPNN structure.

**Figure 7 sensors-22-05515-f007:**
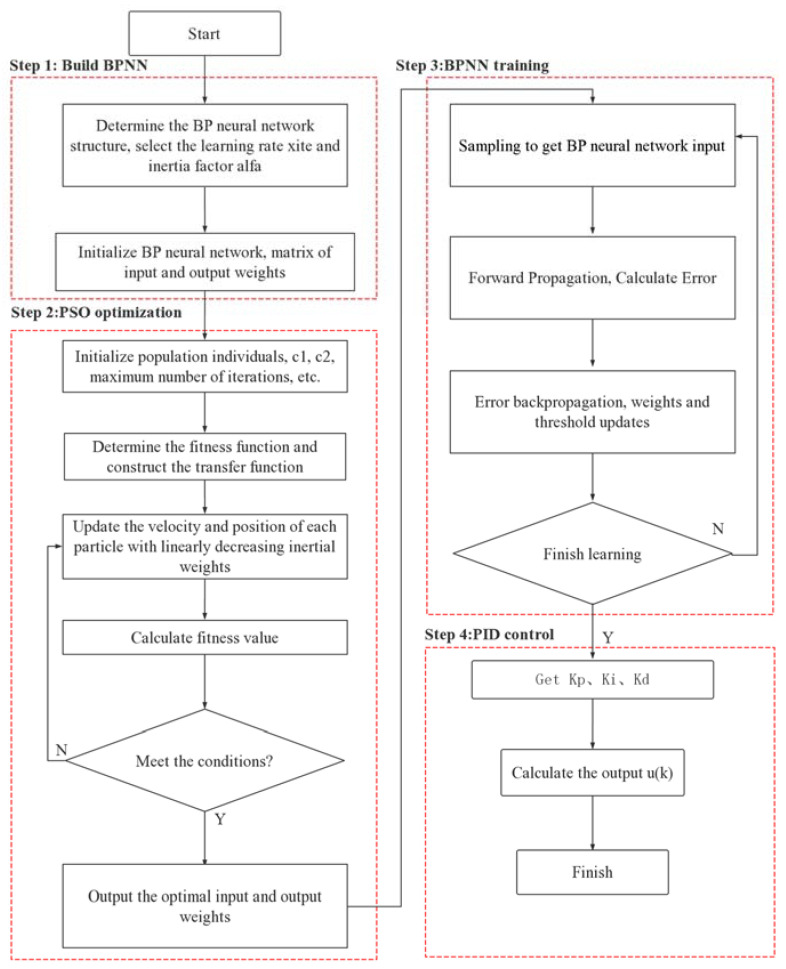
Flow chart of the PSO-BPNN-PID algorithm.

**Figure 8 sensors-22-05515-f008:**
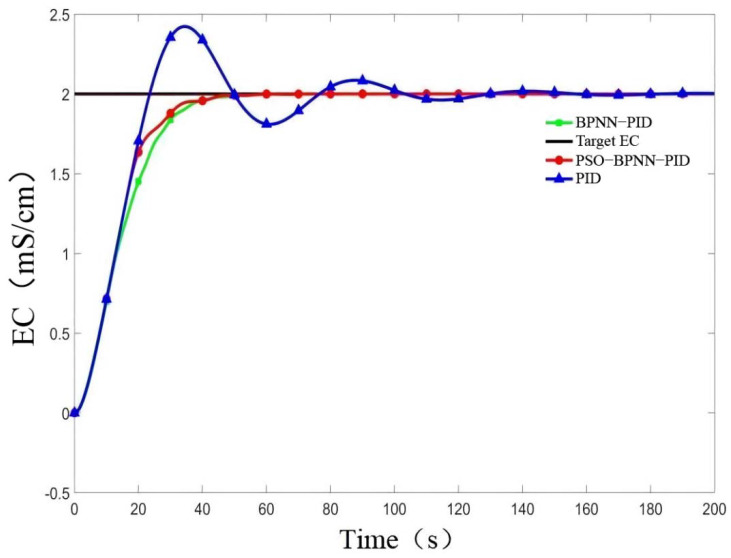
Comparison of different control results.

**Figure 9 sensors-22-05515-f009:**
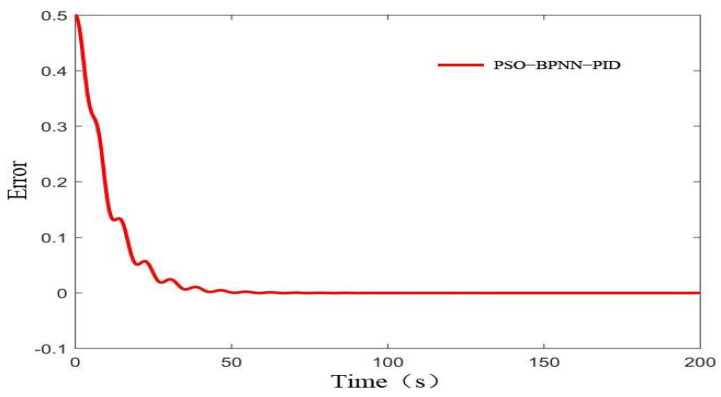
Error curve.

**Figure 10 sensors-22-05515-f010:**
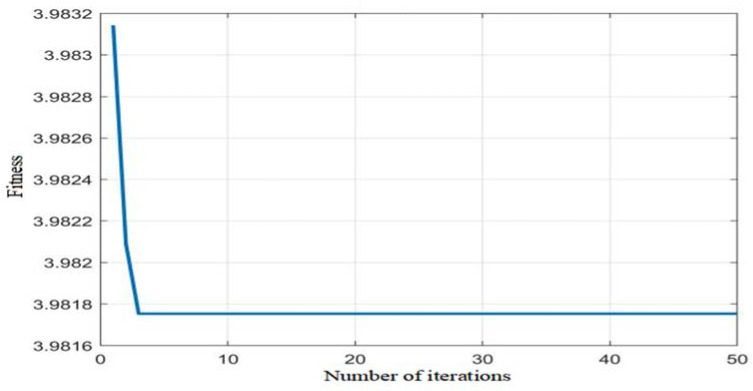
Fitness curve.

**Figure 11 sensors-22-05515-f011:**
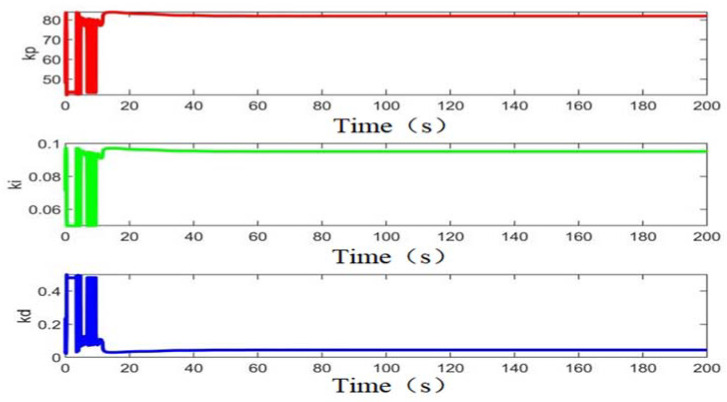
Kp, Ki, and Kd after PSO optimization.

**Figure 12 sensors-22-05515-f012:**
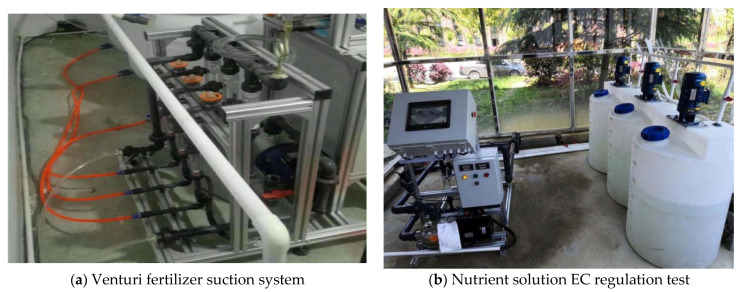
Fertilizer test platform.

**Table 1 sensors-22-05515-t001:** Master node–child node protocol nesting.

Sender		Receiver	
Target group address	0xXXXX	local group address	0x5678
Module channel	0xXXXX	module channel	0x18
Send data	Receive address high + receive address low + receive channel + data (data)	Output Data	User data (data)
	0x56 0x78 0x18 0x11 0x22 0x33		0x11 0x22 0x33

Serial transmission format of user data bytes: 1 start bit; 8 data bits; 1 stop bit, no parity check.

**Table 2 sensors-22-05515-t002:** Data frame format.

Serial Number	1	2	3	4	5	6	7
Number of bytes	1	2	2	2	n	2	1
Code	STA (E8)	AD	C	LEN	DATA	CRC	END (E6)

**Table 3 sensors-22-05515-t003:** Definition of control codes.

**D15**	**D14**	**D13**	**D12**	**D11**	**D10**	**D9**	**D8**
If D15~D8 = 0, it is the communication between the collector and the background; if D15~D8 = 1, it is the communication between the collector and Bluetooth.							
**D7**	**D6**	**D5**	**D4**	**D3**	**D2**	**D1**	**D0**
Transmission direction	Exception flag	Function code					

**Table 4 sensors-22-05515-t004:** Initial weights of the PSO-BP-PID controller.

*wij*				*wki*				
−0.3234	0.3269	−0.3827	−0.0136	0.0208	0.3921	0.0192	−0.4522	−0.0991
0.4135	−0.2866	−0.3405	−0.4651	−0.4179	0.0088	−0.1617	−0.4256	−0.3543
0.2692	0.2864	−0.4887	0.1367	−0.3415	0.1173	0.1547	0.4085	0.3075
−0.1759	0.1077	−0.1479	0.2126					
0.3873	−0.0634	0.3556	0.1271					

**Table 5 sensors-22-05515-t005:** Control parameters and performance indicators for different EC values.

Control Method	Target EC (mS.cm^−1^)	Steady State EC (mS.cm^−1^)	EC Volatility (mS.cm^−1^)	Steady State Time (s)	Overshoot (%)
PSO-BPNN-PID	1	0.9904–1.0003	0.0126	40	0.3
	1.5	1.4990–1.5001	0.0011	45	0.06
	2	1.9997–2.0027	0.0003	43	0.14
	2.5	2.4901–2.5020	0.119	60	0.08

**Table 6 sensors-22-05515-t006:** Close-range test results (within 1 and 100 m).

Frequency	Startup Time/s	Time for Each Issued Command/s	Data Reporting Time/s	Data Receiving Interval/s	Delay/s
1	3.87	1.123	12.156	15.967	3.81
2	3.58	1.124	13.982	15.982	1.50
3	3.78	1.124	12.699	16.699	3.76
4	3.56	1.124	12.835	14.835	1.62
5	3.71	1.124	12.363	15.363	3.02
6	3.87	1.123	12.872	15.872	2.95
7	3.58	1.123	12.635	15.635	3.05
8	3.78	1.122	11.985	14.985	2.47
9	3.56	1.124	12.213	16.213	4.28
10	3.71	1.124	11.987	15.987	3.38
Average value	3.7	1.1235	12.5727	15.7538	2.984

**Table 7 sensors-22-05515-t007:** Long-range test results (1000–1500 m).

Frequency	Startup Time/s	Time for Each Issued Command/s	Data Reporting Time/s	Data Receiving Interval/s	Delay/s
1	3.87	1.123	12.156	20.544	8.39
2	3.58	1.124	13.982	22.563	8.58
3	3.78	1.124	12.699	21.512	8.81
4	3.56	1.124	12.835	20.367	7.53
5	3.71	1.124	12.363	23.762	11.40
6	3.87	1.123	12.872	22.634	9.76
7	3.58	1.123	12.635	20.318	7.68
8	3.78	1.122	11.985	21.024	9.04
9	3.56	1.124	12.213	21.356	9.14
10	3.71	1.124	11.987	20.946	8.96
Average value	3.7	1.1235	12.5727	21.5026	8.929

**Table 8 sensors-22-05515-t008:** Test results of EC regulation system for nutrient solutions.

Fertilizer Intake		EC Value			
Fertilizer Channel	Measured data of Fertilizer Intake(L/h)	Test Number	Target Value (mS/cm)	Measured Value (mS/cm)	Error
1	700.00	1	0.5	0.43	−0.07
2	660.00	2	1	0.92	−0.08
3	675.00	3	1.5	1.65	+0.15
4	738.00	4	2	1.82	−0.18
average value	693.25	5	2.5	2.57	+0.07

## Data Availability

Not applicable.

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
