# Peer review of "Application of PSO-BPNN-PID Controller in Nutrient Solution EC Precise Control System: Applied Research"

_sensors, 2022, doi:10.3390/s22155515_

Round 1

Reviewer 1 Report

With reference to the paper titled “Simulation and experiment of nutrient solution EC precise control system based on PSO-BPNN-PID” (manuscript ID: sensors-1808617), I believe the authors have done an overall good job in terms of contents, but the paper shows shortcomings in terms of presentation quality that must be solved to improve its legibility. The main issues found in the manuscript are the following:

-          The authors are suggested to clarify all the acronyms at the first appearance, also in the Abstract (e.g. EC-Electrical Conductivity?).

-          Related to the sentence, “Under the same initial conditions, when the input EC=2mS/cm, the optimized scale factor of the particle swarm optimization BP neural network PID (PSO-BPNN-PID) control method is Kp=81, Ki=0.095, Kd=0.044, the control The….”, Maybe there is something wrong?

-          The authors are suggested to review the whole paper to improve the English language and correct typos.

-          The authors are suggested to add a brief summary of the remainder of the paper structure at the end of the introduction.

-          The authors are suggested to check all figure references; in some cases, they look wrong (Figure 7.12 on page 5).

-          The authors are recommended to describe better the architecture of the sensor network used to acquire the parameters onfield. In the actual state, it seems incomplete and disordered.

-          The authors are suggested to review the conclusion section to summarize the obtained results better.

Author Response

Dear editors and reviewers:

Thank you very much for your comments concerning our manuscript entitled “Application of PSO-BPNN-PID controller in nutrient solution EC precise control system: applied research” (Manuscript ID: sensors-1808617). Your comments are so valuable and helpful for us to correct and improve our manuscript. We have tried our best to revise and improve the manuscript, and given an explanation according to each issue of your report. The revised portion is marked in red in the revised manuscript. The main corrections and responses to your comments are listed in the appendix.

Yours sincerely,

Yongtao Wang, Jian Liu, Rong Li, Xinyu Suo and EnHui Lu

Appendix 1: Our responses to the reviews’ comments

  • The authors are suggested to clarify all the acronyms at the first appearance, also in the Abstract (e.g. EC-Electrical Conductivity?).

Response:Thank you very much for your careful and valuable comments. We apologize for not clarifying all acronyms in the abstract for the first occurrence, and have now made the following changes to the full text.The contents are:

Abstract: In this paper, we present a nutrient solution control system, designing a nutrient solution electrical conductivity (EC) sensing system composed of multiple long range radio (LoRa) slave nodes, narrow-band Internet of Things (NB-IoT) master nodes and a host computer, building a nutrient solution EC control model, and using the particle swarm optimization (PSO) algorithm to optimize the initial weights of a back-propagation neural network (BPNN). In addition, the optimized best weights are put into the BPNN to adjust the proportional–integral–derivative (PID) control parameters Kp, Ki, and Kd so that the system performance index can be optimized. Under the same initial conditions, we input EC=2mS/cm, and use the particle swarm optimization BP neural network PID (PSO-BPNN-PID) to control the EC target value of the nutrient solution. The optimized scale factors were Kp=81, Ki=0.095, and Kd=0.044; the steady state time was about 43 s, the overshoot was about 0.14%, and the EC value was stable at 1.9997 mS/cm–2.0027 mS/cm. Compared with the BP neural network PID (BPNN-PID) and the traditional PID control approach, the results show that PSO-BPNN-PID had a faster response speed and higher accuracy. Furthermore, we input 1 mS/cm, 1.5 mS/cm, 2 mS/cm, and 2.5 mS/cm, respectively, and simulated and verified the PSO-BPNN-PID system model. The results showed that the fluctuation range of EC was 0.003 mS/cm~0.119 mS/cm, the steady-state time was 40s~60s, and the overshoot was 0.3%~0.14%, which can meet the requirements of the rapid and accurate integration of water and fertilizer in agricultural production.

  • Related to the sentence, “Under the same initial conditions, when the input EC=2mS/cm, the optimized scale factor of the particle swarm optimization BP neural network PID (PSO-BPNN-PID) control method is Kp=81, Ki=0.095, Kd=0.044, the control The….”, Maybe there is something wrong?

Response:Thank you very much for your valuable comments. The sentence is not clearly expressed, and has been revised as follows:

Under the same initial conditions, we input EC=2mS/cm, and use the particle swarm optimization BP neural network PID (PSO-BPNN-PID) to control the EC target value of the nutrient solution. The optimized scale factors were Kp=81, Ki=0.095, and Kd=0.044; the steady state time was about 43 s, the overshoot was about 0.14%, and the EC value was stable at 1.9997 mS/cm–2.0027 mS/cm. Compared with the BP neural network PID (BPNN-PID) and the traditional PID control approach, the results show that PSO-BPNN-PID had a faster response speed and higher accuracy.

  • The authors are suggested to review the whole paper to improve the English language and correct typos.

Response:Thank you for your very good suggestion, we have asked a professional English Editing recommended by MDPI to participate in the copy editing and review the whole paper.

  • The authors are suggested to add a brief summary of the remainder of the paper structure at the end of the introduction.

Response:The original introductory summary of the paper is really not enough, especially the nutrient solution EC regulation mainly focuses on fuzzy control and PID, heuristic algorithm combined with PID, etc., which is added and improved as follows:

Precision irrigation and precise fertilization in the agricultural production process can not only improve the yield and quality of agricultural products, but can also effectively solve the environmental problems caused by excessive fertilization, which is in line with the concept of ecological green development. Generally, the suitable nutrient solution EC range for crops is usually between 0.8 mS/cm and 2.5 mS/cm, and the EC error range is ±0.8 mS/cm. However, the regulation process of EC in nutrient solutions has the characteristics of nonlinearity, time delays, and time variations. The traditional PID controller, suitable for a simple linear system, will have a large error in the EC regulation of a nutrient solution. Therefore, a large number of studies on EC regulation of nutrient solutions have been carried in several countries, mainly focusing on combinations of fuzzy control and PID, heuristic algorithms and PID, etc., as follows.

(1) Combination of fuzzy control and PID: Wang Xiaolong designed a fuzzy PID controller to control the precise ratio of water and fertilizer [1] . Zhang Yubin et al. applied fuzzy PID control technology based on the EC value and pH value, and developed a precise water and fertilizer irrigation control system [2]. Wang Haihua et al. [7] adopted a PI and fuzzy subsection control strategy to better solve the lag and instability problems of water and fertilizer EC regulation [3]. Li Li et al. analyzed the situation of an actual closed cultivation system. To meet their requirements, the structure of the nutrient solution to control the secondary mixed fertilizer was designed, and a mathematical model of the dynamic process of the nutrient solution was established. At the same time, a PI control algorithm was also designed, and their testing verified that the steady-state time dimension of the system was 100 s, the overshoot was 3%, and the control performance was excellent [4].

(2) Combination of heuristic algorithm and PID: the BP neural network has strong nonlinear mapping ability and self-adaptive ability, and its self-learning ability can be used to output the optimal PID controller parameter combination corresponding to a certain optimal control, and thus achieve the desired control effect. However, the initial value of the weight of the traditional BP neural network is randomly selected according to experience. This method leads to slow convergence of the network, which makes it easy to fall into the local optimal solution, and the final result has a large degree of instability [4]. Because the particle swarm algorithm has memory, it can transfer the memory of the best position of the particle in the history of the group to other particles, and has the advantages of fewer parameters to be adjusted and a simple structure, so the particle swarm algorithm was selected to weight the BP neural network. Optimization was performed in the design of PSO-BPNN-PID to improve the control effect. In recent years, scholars have carried out a series of studies in this area. Jiang Liu et al. designed a PID controller based on the BP neural network algorithm and analyzed the vehicle dynamics. The simulation results showed that the dynamic performance of the vehicle was effectively improved under different input conditions [5]. In order to reduce the influence of temperature on the micro-gyroscope, Xia Dunzhu et al. proposed a temperature compensation control method. First, a BP (back-propagation) neural network and polynomial fitting were used to build the temperature model of the micro-gyroscope. Considering the requirements of simplicity and real-time performance, piecewise polynomial fitting was adopted in the temperature compensation system. Then, an integral-separated PID temperature control system was adopted, with a proportional–integral–derivative control algorithm to stabilize the internal temperature of the micro-gyroscope and achieve its optimum performance. The experimental results showed that the combined temperature compensation and control method of the micro-gyroscope could be realized effectively in a prototype of the micro-micro-gyroscope [6]. Alex Alexandridis et al. proposed a new approach to controlling the general properties of nonlinear systems, using an inverse radial basis function neural network model that was able to combine disparate data from a variety of sources. The results revised the ability of the proposed control scheme to process and manipulate a variety of data. Through the data fusion method, it was shown that the method responded in a faster and less oscillatory manner [7]. Jun Wang et al. proposed a closed-loop motion control system based on a BP neural network (BPNN) PID controller, which used a Xilinx field programmable gate array (FPGA) solution. The results showed that the proposed system could realize the self-tuning of PID control parameters, and had the characteristics of reliable performance, high real-time performance, and a strong anti-interference ability. Compared with MCU, the convergence speed was far more than three orders of magnitude, proving its superiority [8]. Yuan Jianping [7] used the GA-PSO-BP-PID algorithm to control the greenhouse environment, and the simulation results obtained using MATLAB showed that the stability and robustness of the control system were better than conventional BP-PID [9]. Li Hang et al. used an improved genetic algorithm to optimize the BP neural network to achieve a better control over the gas concentration [10]. The abovementioned BP neural network research has achieved good results in the field of environmental control, but not in the field of nutrition. Therefore, based on the above research, in this paper we develop a wireless EC sensing system, as well as a nutrient solution EC regulation model, and we further explore the combination of PSO optimization and BP-PID to achieve the precise EC regulation of nutrient solutions. The initial weight of BPNN is continuously optimized by the PSO algorithm to obtain the best weight, and the best weight is input into the BP neural network to automatically adjust the PID control parameters Kp, Ki, and Kd, and the optimal control parameters are obtained. In system simulation and testing, system control performance was found to be excellent, sufficient to meet the needs of actual production.

  • The authors are suggested to check all figure references; in some cases, they look wrong (Figure 7.12 on page 5).

Response:Thank you for your comments which are very important for the improvement of the quality of my paper, and I am sorry for such obvious mistakes due to my lack of care. We have carefully cross-checked the figures, tables, etc. throughout the text.

  • The authors are recommended to describe better the architecture of the sensor network used to acquire the parameters onfield. In the actual state, it seems incomplete and disordered.

Response:Thank you for your very good suggestion. The sensor network architecture for obtaining parameters in the field includes multiple LoRa slave nodes, a master node, a server, and a remote-control terminal. For better illustration, we have updated the block diagram of Figure 2 master and slave nodes.

  • The authors are suggested to review the conclusion section to summarize the obtained results better.

Response:Thanks to your excellent suggestions, we have revisited and summarized the conclusions, highlighting the sensor network architecture and other key findings of the paper, as shown in the following details:

  • In this study, we built a precise EC control system platform for nutrient solutions, and developed an EC information perception system composed of multiple LoRa slave nodes, NB-IoT master nodes, and a host computer. The system startup time was about 3.56~3.87 s; and the command time was approximately 1.1235s. The command time was thus in the range of 1.122~1.124 s, and the data delay was about 1.50~11.4 s. The system startup time, command issuing time, data reporting time, and data receiving time intervals were all relatively fast, and the delay in issuing commands was small, and the data could be reported according to the time required by the user. The system can maintain good running performance.

(2) Based on the PSO optimization method, a BP-PID nutrient solution EC control model with a structure of 4-5-3 was constructed. Under the same initial conditions, when the input EC=2 mS/cm, the PSO-BPNN-PID method was used to control the EC value of the nutrient solution. The optimized PID scaling factors were Kp=81, Ki=0.095, and Kd=0.044; the control steady state time was about 43 s; the overshoot was about 0.14%; and the system’s EC value was stable at 1.9997 mS/cm–2.0027mS/cm. Compared with BPNN-PID and the traditional PID, we achieved steady-state time savings of 16.2% and 67%; the overshoot was reduced by 99.3% and 7.1%; and the system control performance was excellent.

Reviewer 2 Report

>According to my opinion, the abstract should contain the presentation of the paper. Here, details of results are shown (it can be analyzed in other section). It should be modified.

>The quality of the figures should be improved.

>How was the number of hidden neurons exactly selected (section 4.2)? I have noticed a short note. However it needs additional explanations.

>I understand that after training (using the BP method) the weights are fixed. It should be clarified.

>Additional details of algorithms implementation (NN, BP, and PSO) should be presented.

>The introduction can be extended with the following information related to combination of neural networks and heuristic algorithms.

>Have you assumed fair-conditions (initial values, PID controller optimization, network structure, etc.) for comparison (figure 8)?

>Please improve the format of the formulas.

>How were the parameters of the transfer function of the system identified?

>I think, the title of the paper needs minor improvements.

>'this paper is better than the BP neural network PID (BPNN-PID). and the traditional PID' I think, there is a typo.

Author Response

Dear editors and reviewers:

Thank you very much for your comments concerning our manuscript entitled “Application of PSO-BPNN-PID controller in nutrient solution EC precise control system: applied research” (Manuscript ID: sensors-1808617). Your comments are so valuable and helpful for us to correct and improve our manuscript. We have tried our best to revise and improve the manuscript, and given an explanation according to each issue of your report. The revised portion is marked in red in the revised manuscript. The main corrections and responses to your comments are listed in the appendix.

Yours sincerely,

Yongtao Wang, Jian Liu, Rong Li, Xinyu Suo and EnHui Lu

Appendix 1: Our responses to the reviews’ comments

  • According to my opinion, the abstract should contain the presentation of the paper. Here, details of results are shown (it can be analyzed in other section). It should be modified.

Response:Thank you very much for your careful and valuable comments. As you commented, the abstract should contain an introduction to the paper and should not contain detailed information about the results, which We have modified as follows:

Abstract: In this paper, we present a nutrient solution control system, designing a nutrient solution electrical conductivity (EC) sensing system composed of multiple long range radio (LoRa) slave nodes, narrow-band Internet of Things (NB-IoT) master nodes and a host computer, building a nutrient solution EC control model, and using the particle swarm optimization (PSO) algorithm to optimize the initial weights of a back-propagation neural network (BPNN). In addition, the optimized best weights are put into the BPNN to adjust the proportional–integral–derivative (PID) control parameters Kp, Ki, and Kd so that the system performance index can be optimized. Under the same initial conditions, we input EC=2mS/cm, and use the particle swarm optimization BP neural network PID (PSO-BPNN-PID) to control the EC target value of the nutrient solution. The optimized scale factors were Kp=81, Ki=0.095, and Kd=0.044; the steady state time was about 43 s, the overshoot was about 0.14%, and the EC value was stable at 1.9997 mS/cm–2.0027 mS/cm. Compared with the BP neural network PID (BPNN-PID) and the traditional PID control approach, the results show that PSO-BPNN-PID had a faster response speed and higher accuracy. Furthermore, we input 1 mS/cm, 1.5 mS/cm, 2 mS/cm, and 2.5 mS/cm, respectively, and simulated and verified the PSO-BPNN-PID system model. The results showed that the fluctuation range of EC was 0.003 mS/cm~0.119 mS/cm, the steady-state time was 40s~60s, and the overshoot was 0.3%~0.14%, which can meet the requirements of the rapid and accurate integration of water and fertilizer in agricultural production.

  • The quality of the figures should be improved.

Response:Thank you very much for your valuable comments. We have redrawn the paper to the best of my ability with high quality and made replacements in the paper.

  • How was the number of hidden neurons exactly selected (section 4.2)? I have noticed a short note. However it needs additional explanations.

Response:Thank you for your question about how the number of hidden neurons is accurately selected, trapped in space this thesis really did not go into detail, in fact, the method determined in this thesis is: first of all, the number of hidden neurons is initially calculated according to the empirical formula in reference [18]; then, by trial and error analysis of the number of different hidden neurons is performed; finally, the number of hidden neurons is finally determined to be 5, which is modified and added as follows:

Determine the BP neural network structure and select the learning rate xite and inertia factor alfa. The network adopts a three-layer network architecture with one input layer, one hidden layer, and one output layer. The number of nodes in the input layer of the BP neural network is determined by trial and error. The input vector x = [x1, x2, x3, x4], where x1=ek-e1; x2=ek; x3=ek-2·e1+e2; x4=1. The number of hidden layer nodes is h=5, and the output layer O=3, corresponding to the outputs kp, ki, and kd. Therefore, the neural network structure constructed in this paper was 4-5-3 [19–20].

  • I understand that after training (using the BP method) the weights are fixed. It should be clarified.

Response:We strongly agree with you because you say that the trained neural network weights are fixed. In this paper, we optimize the initial weights of the neural network by PSO algorithm BP and input the PSO optimized initial weights into the BP neural network before the BP neural network starts training to obtain the optimal control parameters. The above process is used to determine the optimal weights before the training of the BP neural network.

  • Additional details of algorithms implementation (NN, BP, and PSO) should be presented.

Response:Thank you for your suggestion, the details of the BPNN, PSO optimization algorithm are really important. Most of the space is devoted to it in this paper. The textual description of the 4 steps from Step1: Established BP neural network, Step2: PSO optimization, Step3: BP neural network training, PID control, and combined with the Figure 5. PSO-BP-PID controller structure, Figure 6. Nutrient EC control BPNN structure, Figure 7. Flow chart of the PSO-BPNN-PID algorithm, etc. are described in detail with additional graphics.

  • The introduction can be extended with the following information related to combination of neural networks and heuristic algorithms.

Response:Thank you for your very good suggestion, the information related to the combination of neural networks and heuristic algorithms has been extended in the introduction of this thesis with the following details:

Combination of heuristic algorithm and PID: the BP neural network has strong nonlinear mapping ability and self-adaptive ability, and its self-learning ability can be used to output the optimal PID controller parameter combination corresponding to a certain optimal control, and thus achieve the desired control effect. However, the initial value of the weight of the traditional BP neural network is randomly selected according to experience. This method leads to slow convergence of the network, which makes it easy to fall into the local optimal solution, and the final result has a large degree of instability [4]. Because the particle swarm algorithm has memory, it can transfer the memory of the best position of the particle in the history of the group to other particles, and has the advantages of fewer parameters to be adjusted and a simple structure, so the particle swarm algorithm was selected to weight the BP neural network. Optimization was performed in the design of PSO-BPNN-PID to improve the control effect. In recent years, scholars have carried out a series of studies in this area. Jiang Liu et al. designed a PID controller based on the BP neural network algorithm and analyzed the vehicle dynamics. The simulation results showed that the dynamic performance of the vehicle was effectively improved under different input conditions [5]. In order to reduce the influence of temperature on the micro-gyroscope, Xia Dunzhu et al. proposed a temperature compensation control method. First, a BP (back-propagation) neural network and polynomial fitting were used to build the temperature model of the micro-gyroscope. Considering the requirements of simplicity and real-time performance, piecewise polynomial fitting was adopted in the temperature compensation system. Then, an integral-separated PID temperature control system was adopted, with a proportional–integral–derivative control algorithm to stabilize the internal temperature of the micro-gyroscope and achieve its optimum performance. The experimental results showed that the combined temperature compensation and control method of the micro-gyroscope could be realized effectively in a prototype of the micro-micro-gyroscope [6]. Alex Alexandridis et al. proposed a new approach to controlling the general properties of nonlinear systems, using an inverse radial basis function neural network model that was able to combine disparate data from a variety of sources. The results revised the ability of the proposed control scheme to process and manipulate a variety of data. Through the data fusion method, it was shown that the method responded in a faster and less oscillatory manner [7]. Jun Wang et al. proposed a closed-loop motion control system based on a BP neural network (BPNN) PID controller, which used a Xilinx field programmable gate array (FPGA) solution. The results showed that the proposed system could realize the self-tuning of PID control parameters, and had the characteristics of reliable performance, high real-time performance, and a strong anti-interference ability. Compared with MCU, the convergence speed was far more than three orders of magnitude, proving its superiority [8]. Yuan Jianping [7] used the GA-PSO-BP-PID algorithm to control the greenhouse environment, and the simulation results obtained using MATLAB showed that the stability and robustness of the control system were better than conventional BP-PID [9]. Li Hang et al. used an improved genetic algorithm to optimize the BP neural network to achieve a better control over the gas concentration [10]. The abovementioned BP neural network research has achieved good results in the field of environmental control, but not in the field of nutrition. Therefore, based on the above research, in this paper we develop a wireless EC sensing system, as well as a nutrient solution EC regulation model, and we further explore the combination of PSO optimization and BP-PID to achieve the precise EC regulation of nutrient solutions. The initial weight of BPNN is continuously optimized by the PSO algorithm to obtain the best weight, and the best weight is input into the BP neural network to automatically adjust the PID control parameters Kp, Ki, and Kd, and the optimal control parameters are obtained. In system simulation and testing, system control performance was found to be excellent, sufficient to meet the needs of actual production.

  • Have you assumed fair-conditions (initial values, PID controller optimization, network structure, etc.) for comparison (figure 8)?

Response:Thank you for your very good question, this paper only compares the control effects of different controllers of PSO-BPNN-PID and BPNN-PID-PID under the same conditions set, only then, the results of the comparison adopt meaning and value. This paper is also described in detail in different places, thanks.

  • Please improve the format of the formulas.

Response:Thank you for your very good suggestion, we have re-edited all the formulas in the text according to the editorial template.

  • How were the parameters of the transfer function of the system identified?

Response:Thank you for your very good advice, the transfer function is very important for the construction of the controller. The transfer function in this thesis is determined firstly by initially determining the style of the transfer function according to references [13-15], secondly by calculating the main parameters of the transfer function, and finally, by finalizing and verifying it using the MATLAB system identification method.

  • I think, the title of the paper needs minor improvements.

Response:Thanks for your good suggestion, we have changed the title of the paper to "Application of PSO-BPNN-PID controller in nutrient solution EC precise control system: applied research".

  • This paper is better than the BP neural network PID (BPNN-PID). and the traditional PID' I think, there is a typo.

Response:Thank you for your comments, and we apologize for the editing errors in the text. We have double-checked the figures, tables, and text in the full text.

Round 2

Reviewer 1 Report

The following initial points and requests were not considered and met in the revised version of the manuscript:

  • The authors are suggested to add a brief summary of the remainder of the paper structure at the end of the introduction.
  • The authors are recommended to describe better the architecture of the sensor network used to acquire the parameters onfield. 

More details are necessary concerning the sensor network architecture (sections 2.2.1 and 2.2.2.); instead, the revised version is similar to the initial one. 

Author Response

Dear editors and reviewers:

Thank you very much for your comments concerning our manuscript entitled “Application of PSO-BPNN-PID controller in nutrient solution EC precise control system: applied research” (Manuscript ID: sensors-1808617). Your comments are so valuable and helpful for us to correct and improve our manuscript. We have tried our best to revise and improve the manuscript, and given an explanation according to each issue of your report. The revised portion is marked in red in the revised manuscript. The main corrections and responses to your comments are listed in the appendix.

Yours sincerely,

Yongtao Wang, Jian Liu, Rong Li, Xinyu Suo and EnHui Lu

Appendix 1: Our responses to the reviews’ comments

  • The authors are suggested to add a brief summary of the remainder of the paper structure at the end of the introduction.

Response:This is a good suggestion and for a clearer brief description of the rest of the paper I have included and expanded the nutrient EC control system network structure consisting of sensing, network and application layers, communication performance tests and system control performance tests at the end of the paper as follows:

Therefore, based on the above research, this paper develops an EC control system for nutrient solution consisting of a sensing layer, a network layer and an application layer. The sensing layer consists of multiple LoRa sub-nodes and one LoRa master node, where the master node is connected to the water and fertilizer integration system. By combining LoRa and NB-IoT to form a wireless sensor network, comprehensive sensing, reliable transmission and intelligent application of water and fertilizer control information are realized. The nutrient solution EC regulation model of the water-fertilizer integration system is further constructed, and the PSO-BPNN-PID controller is designed by combining PSO optimization with BP-PID coupling control method, and the initial weights of BPNN are continuously optimized by PSO algorithm to get the optimal weights, and the optimal weights are input into BP neural network to automatically adjust the PID control parameters Kp, Ki and Kd to get the optimal control parameters. After MATLAB simulation with good control effect, further through the self-organized network communication performance test and nutrient solution EC control system test, it is proved that the nutrient solution EC control system has excellent performance and can meet the needs of actual production.

  • The authors are recommended to describe better the architecture of the sensor network used to acquire the parameters onfield. More details are necessary concerning the sensor network architecture (sections 2.2.1 and 2.2.2.); instead, the revised version is similar to the initial one. 

Response:This is a very good suggestion, indeed it is not easy to clearly describe the structure of the sensor network for obtaining field parameters, for this reason, we decided that it is a good choice to use the perception layer, network layer and application layer of the Internet of Things to describe the sensors for obtaining parameters in the field mainly for the perception layer, see the details of the paper (section 3. Design of EC regulation System of Nutrient Solution ) with the following specific details:

  1. Design of EC regulation System of Nutrient Solution

The system can be divided into three parts, consisting of the sensing, network and application layers. The network structure of EC regulation system of nutrient solution is shown in Figure 2.

3.1 Sensing Layer

The sensing layer consists of multiple LoRa slave nodes, and one NB-IoT master node. Each slave node has the ability to collect 4–20 mA, 0–5 V, 485 standard signals of pressure and water level transmitters; to collect flow data from flow meters and pulse meters for cumulative flow calculation; to collect on-site signals such as ambient temperature and humidity, and transmit them through LoRa to the master node.

Each slave node is powered by a 12 V high-performance lithium battery and is powered by a standard power adapter. It has a built-in 5 V voltage regulator chip and a 3.3 V voltage conversion circuit, which can collect the output signals of various instruments and transmitters, and uses LoRa for long distances. It transmits data, controls the operation of a 12 V solenoid valve (switch value), a DC3.6V motor (analog value), collects motor operation status, and transmits data remotely through a NB-IoT signal, which is suitable for monitoring sites without power supply conditions and in harsh environments. Since the node needs to work for a long time in the field, a photovoltaic solar charging device was designed, which has the functions of undervoltage protection, overdischarge protection, overload protection, floating charging settings, overload short circuit protection, overvoltage protection, direct charging voltage, and other functions. Figure 3 shows the structure of the slave master node.

The master node is mainly based on the STM32 controller and is connected to the network through modbus and UDP (User Data Protocol) communication to ensure the reliability of the data and instructions. Through the multi-node coverage and high timeliness of the system, the centralized analysis of data in the irrigation area, as well as remote control and scheduling in the gate cluster, are realized, and it is suitable for the precise control of water delivery and distribution in small and medium-sized channels for farmland irrigation. At the same time, the master node is connected to the water and fertilizer integration system.
